# Socioeconomic Determinants in Vaccine Hesitancy and Vaccine Refusal in Italy

**DOI:** 10.3390/vaccines8020276

**Published:** 2020-06-05

**Authors:** Chiara Bertoncello, Antonio Ferro, Marco Fonzo, Sofia Zanovello, Giuseppina Napoletano, Francesca Russo, Vincenzo Baldo, Silvia Cocchio

**Affiliations:** 1Hygiene and Public Health Unit, Department of Cardiac Thoracic and Vascular Sciences and Public Health, University of Padua, Via Loredan 18, 35131 Padova, Italy; chiara.bertoncello@unipd.it (C.B.); sofia.zanovello@studenti.unipd.it (S.Z.); vincenzo.baldo@unipd.it (V.B.); silvia.cocchio@unipd.it (S.C.); 2Public Health Department, Trento Health Authority, 38123 Trento, Italy; Antonio.Ferro@apss.tn.it; 3Organizational Unit, Prevention and Public Health, Veneto Region, 30123 Venice, Italy; giuseppina.napoletano@aulss9.veneto.it (G.N.); francesca.russo@regione.veneto.it (F.R.)

**Keywords:** vaccine hesitancy, socioeconomic inequalities, childhood vaccination.

## Abstract

Childhood vaccination has been a milestone in the control of infectious diseases. However, even in countries offering equal access to vaccination, a number of vaccine-preventable diseases have re-emerged. Suboptimal vaccination coverage has been called into question. The aim was to explore socioeconomic inequalities in vaccine hesitancy and outright refusal. Families with at least one child aged between 3 months and 7 years were involved through an online survey. Families were classified as provaccine, hesitant, or antivaccine. The association between socioeconomic determinants and hesitancy/refusal was investigated with a logistic-regression model. A total of 3865 questionnaires were collected: 64.0% of families were provaccine, 32.4% hesitant, and 3.6% antivaccine. Rising levels of perceived economic hardship were associated with hesitancy (adjusted odds ratio (AOR) from 1.34 to 1.59), and lower parental education was significantly associated with refusal (AOR from 1.89 to 3.39). Family economic hardship and parental education did not move in parallel. Economic hardship was a determinant of hesitancy. Lower education was a predictor of outright refusal without affecting hesitancy. These findings may serve as warnings, and further explanations of socioeconomic inequities are needed even in universal healthcare systems. Insight into these factors is necessary to improve convenience and remove potential access issues.

## 1. Introduction

Childhood vaccination has been a milestone in the prevention and control of infectious diseases. However, even in high-income countries, a number of vaccine-preventable diseases have re-emerged, being a threat at both the individual and the community level. Suboptimal vaccination coverage in children has been called into question [1]. Thousands of cases involved Europe over the past years: 12,266 cases of measles in 2018, 89 of tetanus in 2017, and 3280 of invasive meningococcal disease in 2016, of which 304 were fatal [2]. Delaying vaccine administration not only extends susceptibility to vaccine-preventable diseases, but also prevents herd immunity to be reached, making it more difficult to protect children who cannot be vaccinated.

Vaccine hesitancy has been defined by the World Health Organization Strategic Advisory Group of Experts (WHO SAGE) working group as the “delay of acceptance or refusal of vaccination despite availability of vaccination services” [3]. However, it is not only a matter of trust in vaccines. As explained in the SAGE model of determinants, vaccine hesitancy is a behavior influenced by a number of factors, not only psychological state or personal beliefs about vaccines. The group acknowledged the high complexity of the phenomenon, and proposed the so-called “3 Cs” model, namely, confidence, complacency, and convenience, to further contextualize factors involved in vaccine hesitancy.

While confidence and complacency predominantly pertain to psychological state, convenience encompasses issues related to vaccine availability, accessibility, and health literacy. Additional causes may also be related to “pragmatics, competing priorities, or the failure of services or policies” [4].

This aspect might still be considered even in countries with publicly funded national vaccination programs that address major financial barriers to vaccination access. The rationale behind universal healthcare systems should be to offer free healthcare for everyone, regardless of their financial condition or personal characteristics. However, socioeconomic inequalities have been described not only in countries without universal health coverage, but also in Canada, Sweden, and Italy, where there are supposedly no financial barriers. In Europe, socioeconomic inequalities in health have not only persisted in the last three decades, but even widened [5]. Inequalities have contributed to increasing the number of susceptible individuals [6], and any delay in vaccination reduces the prospect of reaching herd immunity.

While socioeconomic determinants may have substantial and direct contribution in convenience-related issues, it is difficult to disentangle the complex interactions between confidence and complacency-related issues. Nonetheless, analysis of determinants may serve as a warning light for factors related to access and pragmatics, even in universal healthcare systems.

While the definition of hesitancy is generally used to broadly describe nonvaccination, some refusers are forthright in their positions and probably have never been hesitant [4]. Similarly, there are “hesitant compliers”, i.e., parents that fully vaccinate their children but still have concerns [7]. Vaccine refusal is only a relatively small fraction of the larger issue of vaccine hesitancy, and the border between hesitancy and refusal is definitely blurred [8,9]. However, in a population with suboptimal vaccine-uptake levels, discrimination between vaccine hesitancy and outright refusal may be crucial in the short term to highlight specific hallmarks of two potentially different populations.

Vaccine hesitancy is a dynamic and heterogeneous phenomenon, varying according to context, and to geographic and demographic variables [10]. In Italy, vaccination-uptake levels have been decreasing in recent years, leading to the re-emergence of infectious diseases. Between February 2017 and January 2018, Italy accounted for 34% of cases of measles occurring in the European Union [11], resulting in a call for the national government to introduce compulsory school-entry vaccination [12].

In a recent study, vaccine uptake was defined as a continuum [13]. Past studies conducted in Italy mainly focused on parental opinions regarding childhood vaccination [14,15,16], but socioeconomic determinants may also impact hesitancy. The aim of the present study was to estimate the effect of socioeconomic inequalities on vaccine hesitancy, carefully distinguishing between those people who could be defined as hesitant in the strict sense, and those who completely rejected vaccines as a whole.

## 2. Materials and Methods

In Italy, the national immunization prevention plan is issued by the Ministry of Health. All vaccines included are considered part of the “minimal level of healthcare services” (LEA–Livelli Essenziali di Assistenza) and are actively offered free of charge [17]. Childhood vaccinations are usually administered by vaccination services in local health units [18].

The survey was conducted between December 2016 and April 2017, before the introduction of compulsory vaccination in Italy that happened in June 2017 [12].

The questionnaire was pretested within a group of 20 volunteers to evaluate the clarity and understandability of the proposed items. According to issues raised by the pilot study, minor grammatical and lexical changes were made, but no substantial changes were made. Participant families were informed of the study aim, and consent was inferred by completion of the questionnaire. Data were treated with full confidentiality in accordance with Italian and European legislation. The study complied with the Declaration of Helsinki.

An online questionnaire was administered to families with at least one child aged between 3 months and 7 years. By the age of 3 months, all children should have received at least the first vaccine dose in accordance with the Italian national immunization prevention plan.

In the case of more than one child of the target age, parents had to refer to the youngest. Participation was anonymous and voluntary, without any form of compensation for time.

Information on parental sociodemographic characteristics (age, nationality, education, employment status, employment in healthcare of at least one parent, noncohabiting parents, and perceived family economic hardship), health-related characteristics (self-reported occurrence of adverse events following vaccination in the child at issue, sharing decision making about vaccination between parents), child-related characteristics (age, siblings, prematurity, breastfeeding), and lifestyle-related characteristics (vegetarian–vegan diet, organic-food consumption) were collected.

To assess the role of each parent in decision making, participants were asked to indicate the influence of both mother and father on a 5-point scale ranging from none to very much. Decisions were classified as equally shared if the two answers were equal, and mainly mother or mainly father in all other cases.

Perceived economic hardship was assessed with the question “Is your family experiencing economic hardship? If yes, how would you define it?” Response options were “low”, “moderate”, or “severe”. The definition was borrowed from Italian surveillance system “progressi delle aziende sanitarie per la salute in Italia” (PASSI), a similar tool to the behavioral-risk-factor surveillance system (BRFSS). The PASSI surveillance system is a validated tool and has been a well-established practice since its implementation in 2008 in all Italian regions. It is characterized as a public-health surveillance tool that collects, continuously and through sample surveys, information from the Italian adult population on lifestyles and behavioral risk factors related to the onset of chronic noncommunicable diseases, and the degree of knowledge and adherence to the intervention programs that the country is implementing for their prevention [19].

Families were asked whether they underwent all suggested vaccinations, only some, or none in order to assess vaccine uptake, and if the vaccines were done on time (or postponed by healthcare professionals) in accordance with the recommended schedule issued by the national immunization prevention plan. In those who did not vaccinate their children at all, partially did, or delayed vaccination, their intentions were assessed with the following question: “Are you going to vaccinate your child?” The response options were “yes”, “did not decide yet”, and “absolutely no”. On the basis of the definition of vaccine hesitancy developed by the WHO SAGE working group, an operational definition was developed in line with the definition used by Giambi and colleagues in a previous Italian study [14]. On the basis of self-reported vaccination status, timeliness of vaccinations, and intention, participant families were classified as provaccine, hesitant, or antivaccine. Provaccine families were those who underwent all suggested vaccinations on time (or delayed only because of medical indications) and showed clear positive intentions. Antivaccine families were those who never vaccinated their child and showed clear intentions not to vaccinate in the future. Hesitant families were those who underwent partial or delayed vaccine administration, or did not completely reject the possibility to receive the vaccine.

The association between sociodemographic and health-related characteristics, and hesitancy was investigated with a logistic-regression model (child and lifestyle-related characteristics were assumed as confounders). All variables were entered in a single step. Similar analysis was conducted to assess the association between the same baseline characteristics (except for the occurrence of previous vaccine-adverse events) and vaccine refusal.

The strength of investigated associations was described using adjusted odds ratios (AORs) and corresponding to 95% confidence intervals. Statistical significance for all tests was set at *p* ≤ 0.05 (two-sided). Analyses were performed using IBM SPSS Statistics^®^ version 23 (IBM Corp., Armonk, NY, USA).

## 3. Results

A total of 3865 questionnaires were included in the study: 2474 (64.0%) families were classified as provaccine, 1253 (32.4%) as hesitant, and 138 (3.6%) as antivaccine. Of the families, 23.8% declared not to have economic problems, 48.6%, 23.5% and 4.1% defined their perceived economic hardship as low, moderate, and severe, respectively (Table 1).

While 70.5% of mothers had a higher-education degree, the corresponding proportion among fathers was 48.5%. In most cases (79.2%), parents equally contributed in the decision whether or not to vaccinate their child. In 6.2% of cases, participants reported to have experienced a moderate/severe adverse event after a previous vaccination. Mean ages for mothers and fathers were 37.4 (SD: 4.9) and 40.1 (SD: 5.7), respectively. In 81.0% of the involved families, both parents were employed, and in 20.2% of families, there was at least one parent employed as a healthcare worker.

### 3.1. Determinants of Vaccine Hesitancy

According with results from the logistic-regression analyses (Table 2), rising levels of economic hardship were significantly associated with hesitancy (AOR ranging from 1.34 to 1.59), while no significant association was noted with between parents’ education level and hesitancy.

Although in most of cases, parents equally contributed to decision making, the lack of equal contribution between parents was significantly associated with hesitancy (mainly mother, AOR: 2.82; 95% CI: 2.29–3.46; mainly father, AOR: 3.56, 95% CI: 2.31–5.50). Past experience of adverse events following vaccination was associated with hesitancy, especially in cases of events reported as moderate (AOR: 3.36; 95% CI: 2.41–4.66) or severe (AOR: 8.65; 95%CI: 1.49–50.26). Foreign nationality of parents was not associated with hesitancy. Parental employment status was not associated with hesitancy, nor was the presence in the family of at least one parent working as health professional.

### 3.2. Determinants of Vaccine Refusal

No association was noted between refusal and family economic hardship. On the other hand, lower education among the parents, both mother and father, was significantly associated with vaccine refusal. Compared with mothers holding a degree, those with high-school- and primary-school-level education showed an AOR of 1.89 (95% CI: 1.23–2.93) and 3.39 (95% CI: 1.24–9.28), respectively. In a similar manner, fathers with a high-school education showed an AOR of 1.99 (95% CI: 1.27–3.11), and those with primary-school education an AOR of 2.63 (95% CI: 1.41–4.94) compared to those with the highest education level. As with hesitancy, the lack of equally shared decision making between parents was associated with higher chances of refusal (mainly mother, AOR: 1.67, 95% CI: 1.02–2.74; mainly father, AOR: 3.18 95% CI: 1.27–7.99). The nationality of parents and their employment status were not associated with vaccine refusal, nor was the case of at least one parent working as healthcare professional.

## 4. Discussion

Suboptimal vaccine coverage is often associated with socioeconomic inequities [20]. Underlying causes are context-dependent, and vaccine hesitancy may result from a complex interaction between behavioral and social factors. For example, where hesitancy was assumed to be the root cause of low uptake, closer study disclosed the role of other factors such as delivery of vaccinations [21].

### 4.1. Economic Hardship and Education Level 

Our study revealed how economic hardship represents a determinant of vaccine hesitancy, while no association was found between economic hardship and vaccine refusal. On the other hand, the lower education of both mother and father was a valid predictor of refusal of all vaccines, while hesitancy seemed to not be affected by parental education. The decision not to take education alone as a proxy for socioeconomic status [22] was shown. Our study showed that education and economic hardship do not necessarily move in parallel with respect to vaccine hesitancy, as reported by Feiring and colleagues with respect to human papillomavirus (HPV) vaccination [23].

The specific association between vaccine hesitancy and family economic hardship offers an opportunity to investigate roots of hesitancy that may be related with convenience [4]. Especially among hesitant parents, factors other than personal beliefs or pure skepticism towards vaccines may tip the balance in favor of vaccination. Actual accessibility may affect vaccine uptake, as stated by Danis and colleagues, who identified long distance as a barrier to vaccination [22]. Similarly, vaccination campaigns in schools, universities, and workplaces were effective in increasing vaccine coverage [21], and pharmacy-based interventions have started to be considered, although evidence on the effectiveness in increasing vaccine uptake is uncertain, and a legal framework is missing in most countries [24]. In our context, vaccines were free of charge. However, the accessibility issue should still be considered. While some parents might still value vaccines, vaccination may be a low priority due to domestic or work pressure [4]. Disadvantaged children may, therefore, face a variety of logistical challenges in accessing vaccination services: for example, the lack of money for transportation or a practical difficulty in reconciling vaccination-service opening hours with parents’ working time. Recently, the implementation of a vaccine service with extended opening hours in a disadvantaged suburb in southern Italy led to an increase of up to 25% in vaccine coverage for the MMR (measles, mumps and rubella) vaccine [25]. Initiatives like this could be considered more extensively because they appear to be effective in facilitating access to vaccines in the “last mile”, where no tools are generally provided even in the standard offerings of classical universal healthcare systems.

Our findings showed that a higher level of education seemed to be a protective factor against refusing vaccines. However, there was no consensus about this association in other studies, some being in contrast [26,27], in accordance [1,23,28,29], or showing no significant association [20]. Parents with a higher-education background may use selected sources of information, relying on a critical-thinking attitude and making more active choices [23]. According to Special Eurobarometer 488, individuals with higher education are more likely to acknowledge the danger of vaccine-preventable diseases and the effectiveness of vaccines [30], although highly educated parents have been earlier reported to have more negative attitudes towards vaccines [27]. In our study, accessibility-related issues seem to not have affected the fraction of vaccine refusal, where commitment to a belief is probably total and affected by education.

### 4.2. Effect of Equally Shared Decision-Making Process between Parents

The process of decision making about childhood vaccination has usually been described as “parental”, but it is often reflected only by the maternal position [31]. In our study, both the mother’s and father’s influence on the decision were considered. In many cases, parents equally contributed to decision making, and lack of sharing was associated with hesitancy and refusal. In a previous Italian study, no significant association was found [14]. This finding suggested a potential benefit of involving both parents in consulting activities, although this is not always easy to achieve due to parents’ family and work commitments.

### 4.3. Role of Past Experience of Adverse Events Following Previous Vaccine Administration

Understandably, parents may fear vaccines if they have experienced an adverse event, and they probably would not accept further vaccination. However, a previous Italian study showed that this concern may be overcome, since the implementation of a dedicated vaccine service addressed to children who experienced previous adverse reactions led to an increase in immunization rates [32]. Similar initiatives should be encouraged, and impact may be high, considering that past experiences of adverse events in our study was one of the strongest predictors of hesitancy. However, because of self-reported information, it was not possible to exclude a reverse association in which the hesitant family was more prone to overestimating the severity of the reported adverse events.

Interestingly, and consistent with Italian and international studies, parents working as healthcare workers were as hesitant as other parents [15,33,34].

A strength of this cross-sectional study was the use of individual data on socioeconomic factors, and not the use of aggregates of socioeconomic conditions such as deprivation indices based on the postcodes of the residences of participants. The sample was large in size and provided a good cross-section of the population, although selection bias cannot be excluded since families with no Internet access or computer literacy could not take part in the survey. However, it is fair to assume that the size of this bias was very small by virtue of the fact that Italian families with at least one minor child have Internet access in 96.3% of cases [35]. All information was self-reported by participants, and social desirability bias may not be excluded. More accurate information about vaccination status may have been inferred by resorting to official immunization records.

## 5. Conclusions

In conclusion, delaying vaccine administration not only extends susceptibility to vaccine-preventable diseases, but also increases the chances of not completing the recommended vaccination schedule. Moreover, this prevents herd immunity from being reached, making it more difficult to protect children who cannot be vaccinated. Our study revealed how family economic hardship represented a determinant of vaccine hesitancy, while no association was found between economic hardship and vaccine refusal. On the other hand, the low education of both mother and father was a valid predictor of the outright refusal of all vaccines, while hesitancy seemed to not be affected by parental education. To the best of our knowledge, no study considered maternal education, paternal education, and family economic hardship at the same time. Our findings may serve as preliminary considerations, since determinants of hesitancy were shown to be country- and context-specific, although some commonalities between countries can be found. Our study aimed to separately investigate education and economic difficulties, and this would add a further element of comparison between and within countries. Our results were partially unexpected in a setting with presumably equal access to vaccination. This suggests the need for further explanation of the factors underlying the socioeconomic inequities that we observed, focusing on the different roles of parental education and family economic hardship. Insight into these factors is necessary to tailor interventions aimed at improving vaccine convenience, extending accessibility, and ultimately reducing socioeconomic inequities.

## Figures and Tables

**Table 1 vaccines-08-00276-t001:** Sociodemographic and health-related characteristics of study population.

Characteristics of the Sample		*n*	%
Perceived family economic hardship	None	919	23.8%
	Low	1880	48.6%
	Moderate	907	23.5%
	Severe	159	4.1%
Mother’s education *	Degree	2721	70.5%
	High school	1057	27.4%
	Primary school	84	2.2%
Father’s education *	Degree	1865	48.5%
	High school	1600	41.6%
	Primary school	378	9.8%
Parents’ decision making about vaccination	Equally shared	3063	79.2%
	Mainly mother	677	17.5%
	Mainly father	125	3.2%
Adverse events following vaccination (self-reported) *	None	1324	39.8%
	Mild	1799	54.0%
	Moderate	200	6.0%
	Severe	7	0.2%
Mother’s age (years; mean, SD)		37.4	4.9
Father’s age (years; mean, SD)		40.1	5.7
Foreign mother		83	2.1%
Foreign father		83	2.2%
Employment status *	Both parents employed	3105	81.0%
	Only father employed	653	17.0%
	Only mother employed	77	2.0%
Parent employed in healthcare (at least one)		774	20.2%
Noncohabiting parents		82	2.1%
Total		3865	100.0%

* Column totals may not add to total due to missing data in variable at issue.

**Table 2 vaccines-08-00276-t002:** Logistic regression. Determinants of vaccine hesitancy and outright vaccine refusal among parents. AOR = adjusted odds ratio.

Determinants		Hesitant Parents			Antivaccine Parents		
		AOR	95% CI		AOR	95% CI	
	Ref. Provaccine parents						
Perceived family economic hardship	None						
Low		1.341	1.080	1.665	1.182	0.721	1.936
Moderate		1.421	1.100	1.837	1.255	0.717	2.197
Severe		1.590	1.001	2.525	0.863	0.310	2.404
Mother’s education	Degree						
High school		0.845	0.683	1.045	1.895	1.226	2.929
Primary school		0.867	0.442	1.697	3.395	1.241	9.284
Father’s education	Degree						
High school		1.204	0.999	1.451	1.991	1.275	3.107
Primary school		0.932	0.669	1.299	2.635	1.406	4.936
Parents’ decision making about vaccination	Equally shared						
Mainly mother		2.816	2.289	3.465	1.674	1.022	2.742
Mainly father		3.564	2.307	5.505	3.182	1.266	7.997
Adverse events following vaccination (self-reported)	None						
Mild		1.192	0.996	1.426	-	-	-
Moderate		3.356	2.417	4.660	-	-	-
Severe		8.649	1.489	50.256	-	-	-
Mother’s age (years)		0.988	0.963	1.014	0.997	0.944	1.053
Father’s age (years)		1.014	0.994	1.035	1.005	0.962	1.049
Foreign mother		0.593	0.309	1.139	0.582	0.113	2.994
Foreign father		1.137	0.634	2.039	1.427	0.399	5.106
Employment status	Both parents employed						
Only father employed		0.984	0.774	1.250	0.988	0.598	1.631
Only mother employed		0.942	0.509	1.745	0.375	0.049	2.883
Parent employed in healthcare (at least one)		0.995	0.802	1.234	1.315	0.815	2.124
Noncohabiting parents		1.067	0.579	1.965	0.325	0.040	2.641

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
