# Peer review of "Socioeconomic Determinants in Vaccine Hesitancy and Vaccine Refusal in Italy"

_vaccines, 2020, doi:10.3390/vaccines8020276_

Round 1

Reviewer 1 Report

This study proposed by Bertoncello et al tries to decipher how socioeconomic determinants could affect decision/hesitancy to vaccinate or not. The story is pretty well written. Alternatively it seems really too light at this stage to really appreciate the interest of such study and to be suitable for publication in Vaccines.

Hope these comments will help to improve the study:

1-Since the study is based on voluntary participation. Thus, it sounds particularly difficult to ensure that the whole study is not biased at numerous levels. Notably, only people able to read can reply excluding thus the less educated people. Alternatively, the most educated have generally no time to reply to such questionnaire, excluding thus the most educated. This should be better discussed or studied

2-It is not clear whether people who replied are living in cities or coming from any place since it completely influences the decision to vaccinate as well. This should be taken into account

3-How people who replied geographically distributed in Italy? North, South or equally distributed? This could be of interest.

4- Do the authors think that such element affecting decision to vaccinate or not in Italy could be extended to other countries in Europe?

5-What is the percentage of mono or multi-parental (recomposed family) groups? This could impact the involvement of the parents and thus the delay in making the decision

6-Taking into account the high percentage of people receiving additional not declared income it sounds very difficult to really appreciate whether or not people are really telling the truth or not. Without system to verify this point it seems very difficult to conclude on the impact of the economic hardship.

Author Response

Point-by-point response.

This study proposed by Bertoncello et al tries to decipher how socioeconomic determinants could affect decision/hesitancy to vaccinate or not. The story is pretty well written. Alternatively it seems really too light at this stage to really appreciate the interest of such study and to be suitable for publication in Vaccines.

Hope these comments will help to improve the study:

1-Since the study is based on voluntary participation. Thus, it sounds particularly difficult to ensure that the whole study is not biased at numerous levels. Notably, only people able to read can reply excluding thus the less educated people. Alternatively, the most educated have generally no time to reply to such questionnaire, excluding thus the most educated. This should be better discussed or studied

ANSWER: Thank you for your suggestion. Definitely, participation on a voluntary basis poses a number of potential bias that we tried to address in the limitation section of our manuscript. In this particular case, we considered the proportion of population with internet access and we included this finding in the discussion. As regards illiteracy, the proportion of Italian population in that age range stands at 0.37% [http://dati-censimentopopolazione.istat.it/Index.aspx?DataSetCode=DICA_GRADOISTR1], so that we preferred not to include this further consideration in the manuscript – in order to improve readability.

2-It is not clear whether people who replied are living in cities or coming from any place since it completely influences the decision to vaccinate as well. This should be taken into account

ANSWER: People experiencing economic hardship live both in urban and rural areas. In the manuscript we mentioned as example a public health intervention in a suburb with relevant accessibility issues. Variables known to affect hesitancy have been included in our adjusted model.

3-How people who replied geographically distributed in Italy? North, South or equally distributed? This could be of interest.

ANSWER: The sample provides a good cross-section of the population from a geographical point of view (northwest: 32%; northeast: 27%; center: 31%; south: 10%).

4- Do the authors think that such element affecting decision to vaccinate or not in Italy could be extended to other countries in Europe?

ANSWER: Findings from our study may be relevant for countries other European countries other than Italy, but also out of Europe, especially where there are similarities in the health systems (as stated in our manuscript). However, our results may serve as a preliminary consideration, since determinants of hesitancy has been shown to be country- and context-specific, although some commonalities between countries can be found, such as the presence of vaccine-hesitant healthcare providers and concerns about vaccine safety and utility (Publications Office of the European Union. Let’s talk about hesitancy: enhancing confidence in vaccination and uptake: practical guide for public health programme managers and communicators. 2016. https://op.europa.eu/en/publication-detail/-/publication/ebd475ae-4a4f-11e6-9c64-01aa75ed71a1). Our study aims to investigate education and economic difficulties separately and this would add a further element of comparison between and within countries.

5-What is the percentage of mono or multi-parental (recomposed family) groups? This could impact the involvement of the parents and thus the delay in making the decision

ANSWER: Thank you for your consideration. Regardless the composition of the family sharing the same household, we considered the involvement of both parents (not directly related with cohabitation) in the decision-making to be more relevant.

6-Taking into account the high percentage of people receiving additional not declared income it sounds very difficult to really appreciate whether or not people are really telling the truth or not. Without system to verify this point it seems very difficult to conclude on the impact of the economic hardship.

ANSWER: Our question was based on a mere perception on family wealth and there was no need to directly refer to family income. Moreover, the participation was completely anonymous and it seems reasonable to assume that there is no specific reason why answers might be biased in this respect. In any case, the questionnaire used – to assess, among the other variables, the economic hardship – has been validated at a national and international level and it is well established practice. In agreement with your suggestion, this aspect has been described further in the methods.

Reviewer 2 Report

Congratulations on this important study. In fact, it is, in my opinion, a good article, and my little suggestions were just concerning to formatting:

I suggest you change the color of the titles of table 1 (line 147) and table 2 (line 163)  from red to black.

Discussion is very well organized. 

Usually, in conclusion, we don't present bibliographic references. take off the bibliographic references from the conclusion - Usually, in the conclusion, we don't present bibliographic references"

Don’t have nothing to point on the methodology description and the discussion of the results.

The theoretical  approach it is also good.

Author Response

Thank you for your suggestions and appreciation.

Tables have been edited and formatted according to your comments. Moreover, conclusions have been further extended and the bibliographic reference has been removed. All changes made in the manuscript have been highlighted in green.

Round 2

Reviewer 1 Report

Responses to former concerns are pretty clear and modification quite convincing. While not absolutely perfect in the design because it is difficult to evaluate the diversity of groups voluntarily replying to the questionnary, obtained results show at least an overview of the opinion of the most representative part of people. The manuscript is suitable for publication.